# Comparison of Phenolic Contents and Scavenging Activities of Miang Extracts Derived from Filamentous and Non-Filamentous Fungi-Based Fermentation Processes

**DOI:** 10.3390/antiox10071144

**Published:** 2021-07-19

**Authors:** Aliyu Dantani Abdullahi, Pratthana Kodchasee, Kridsada Unban, Thanawat Pattananandecha, Chalermpong Saenjum, Apinun Kanpiengjai, Kalidas Shetty, Chartchai Khanongnuch

**Affiliations:** 1Interdisciplinary Program in Biotechnology, The Graduate School, Chiang Mai University, Mueang, Chiang Mai 50200, Thailand; abdullahialiyud.ada@gmail.com (A.D.A.); ichineung@gmail.com (P.K.); 2Division of Biotechnology, School of Agro-Industry, Faculty of Agro-Industry, Chiang Mai University, Mueang, Chiang Mai 50100, Thailand; kridsada_u@cmu.ac.th; 3Research Center for Multidisciplinary Approaches to Miang, Chiang Mai University, Mueang, Chiang Mai 50200, Thailand; apinun.k@cmu.ac.th; 4Department of Pharmaceutical Sciences, Faculty of Pharmacy, Chiang Mai University, Mueang, Chiang Mai 50200, Thailand; thanawat.pdecha@gmail.com (T.P.); chalermpong.saenjum@gmail.com (C.S.); 5Division of Biochemistry and Biochemical Technology, Department of Chemistry, Faculty of Science, Chiang Mai University, Mueang, Chiang Mai 50200, Thailand; 6Global Institute of Food Security and International Agriculture (GIFSIA), Department of Plant Sciences, North Dakota State University, Fargo, ND 58108, USA; kalidas.shetty@ndsu.edu

**Keywords:** Miang extracts, phenolic contents, antioxidant, fermented tea

## Abstract

The study investigated the impact of the fermentation process on the phenolic contents and antioxidant and anti-inflammatory activities in extracts of Miang, an ethnic fermented tea product of northern Thailand. The acetone (80%) extraction of Miang samples fermented by a non-filamentous fungi-based process (NFP) and filamentous fungi-based process (FFP) had elevated levels of total polyphenols, total tannins, and condensed tannins compared to young and mature tea leaves. The antioxidant studies also showed better the half-maximal inhibitory concentration (IC_50_) values for fermented leaves in both 1,1-diphenyl-2-picrylhydrazyl (DPPH) and 2,2′-azino-bis (3-ethylbenzothiazoline-6-sulfonic acid) (ABTS) radical scavenging activity assays as well as improved ferric reducing antioxidant power (FRAP) compared to young and mature tea leaves. Extracts of NFP and FFP samples at concentrations of 50 and 100 ppm showed better protective effects against hydrogen peroxide (H_2_O_2_)-induced intracellular reactive oxygen species (ROS) production in HT-29 colorectal cells without exerting cytotoxicity. Additionally, lipopolysaccharide (LPS)-induced production of nitric oxide (a proinflammatory mediator as well as a reactive nitrogen species) was also inhibited by these fermented Miang extracts with an IC_50_ values of 17.15 μg/mL (NFP), 20.17 μg/mL (FFP), 33.96 μg/mL (young tea leaves), and 31.33 μg/mL (mature tea leaves). Therefore, both NFP-Miang and FFP-Miang showed the potential to be targeted as natural bioactive functional ingredients with preventive properties against free radical and inflammatory-mediated diseases.

## 1. Introduction

Tea (*Camellia sinensis*) ranks high on the list of most consumed popular beverages in the world and is proven to possess health benefits, viz. antimicrobial, antioxidant, anti-inflammatory, antitumor, antihypertensive, anti-obesity, and antidiabetic properties [1,2,3]. Fermentation has been strongly correlated with an increased bioactivity of tea leaves as has been reported for the microbial fermented Chinese Pu-erh tea, Kombucha tea, and others [4,5]. Miang is an ethnic fermented tea (*Camellia sinensis* var. *assamica*) held in high socioeconomic and cultural esteem in northern Thailand [6]. Depending on the type of fermentation process employed, Miang can be produced either via filamentous fungal and bacterial (predominantly *Lactobacillus* spp.) stepwise fermentation or through exclusive bacterial fermentation under anaerobic conditions (Figure 1). These fermentation types, viz. filamentous fungi growth-based process (FFP) and the non-filamentous fungi growth-based process (NFP), have found their niches in the eastern (including Chiang Rai, Phayao, Lampang, Phrae, and Nan provinces) and western (Chiang Mai, Lumphun, and Mae Hong Son provinces) regions of northern Thailand (ancient eastern and western Lanna Kingdom) [6]. Kodchasee et al. [7] and Unban et al. [8] have extensively investigated and simulated FFP and NFP fermentation over an extended period using fresh tea leaves sourced from Mae-lua village, Phrae Province, and Papae district, Chiang Mai Province, Thailand respectively. These sampling areas have been established to produce significant concentrations of bioactive compounds in Miang [9] as well as possess well acceptable product quality compared to Miang from other plantation areas (unpublished data). In addition, their studies reported the predominant microbial communities in FFP-Miang and NFP-Miang, various organic acids emanating due to microbial metabolism, phenolic content, and DPPH radical scavenging activities of Miang samples.

Although the type of microbial fermented tea and cultural diversities amongst consumers are different; these processes initiate biotransformation and/or oxidation of monomeric phenolics such as green tea catechins and flavonoid aglycones into oxidized phenolics, which include theaflavins, thearubigins, and flavonoid glycosides, which more than compensate for mild/serious catechin depletion by increasing its bioactivity compared to unfermented tea leaves [10,11]. The extent of this biotransformation depends on the fermentation process employed, microbes involved, and many other factors [12]. Moreover, even though the presence of these oxidized phenolics is not yet reported in Miang, the fermented tea has shown excellent in vitro antioxidant activities [9]. Unlike Pu-erh, Kombucha, Zijuan, Qingbrick, Fuzhuan, Oolong, and others whose antioxidative, anti-inflammatory and anticancer effects have already been explored and, thus reported in the literature, there is a paucity of information regarding the modulatory and inhibitory effects of Miang in a diseased cell-based study. In this regard, there is a report by Yang et al. [13] that showed the phosphate-buffered saline (PBS) extract of Miang contained factors that interfered with DNA synthesis via the inhibition of phytohemagglutinin-induced thymidine incorporation in rat mammary tumor and mouse lymphocyte cells, attaining a 72 h inhibition time when treatment was done at experimentally relevant extract concentrations, thus inhibiting tumor growth.

The present research compared the phenolic content, antioxidant, and potential anti-inflammatory effects of Miang in vitro and in a cell-based studies. This will provide initial insights as to how different fermentation processes impact chemical properties as well as biological activities of fermented Miang samples while also providing a clear contrast of the activities of these extracts in relation to their fresh tea leaves (starting material for fermentation).

## 2. Materials and Methods

### 2.1. Sample Collection

NFP-based and FFP-based Miang samples were obtained from two local producers in Papae sub-district, Mae-Tang district, Chiang Mai Province and Pa-dang sub-district, Meuang district, Phrae Province, Thailand, respectively. The starting raw materials for NFP and FFP fermentation were fresh young tea leaves (comprised of the second to the sixth leaf from the tea shoot), and mature tea leaves (the sixth leaf from the tea shoot) were also collected from the locations mentioned above.

### 2.2. Preparation of Tea Extracts

Fresh young and mature tea leaves, raw materials for NFP-Miang and FFP-Miang, respectively, were dried at 50 °C for 24 h in a vacuum dryer (VD53 Binder Oven, Germany) and ground into powdered forms using a blender. To each sample, 80% acetone was added in a sample–solvent ratio 1:20 and extracted in a shaking incubator (Daihan Labtech Co., Ltd., Namyangju-City, Korea) set at 150 rpm, 30 °C for 1 h. The obtained solution was filtered and evaporated using a rotary evaporator (N-1000 Eyela Rotary Evaporator; Tokyo, Japan) at 40 °C, concentrated, and dissolved in deionized (DI) water (RCI Labscan Co., Ltd., Bangkok, Thailand).

### 2.3. Analysis of the Chemical Composition

#### 2.3.1. Total Polyphenol Content

The total polyphenol content was determined according to the Folin–Denis method as described by Padma et al. [14] with some modification. Briefly, 250 μL of sample was made up to 1875 μL by adding 1625 μL of DI water. Then, 125 μL of Folin–Denis reagent (2 M) was added to the mixture and vortexed (Vortex Genie 2, Scientific Industries, Bohemia, NY, USA). Next, 250 μL of sodium carbonate (10%, *w*/*v*) was dispensed into the mixture followed by a second vortex. The sample solution was adjusted to 2.5 mL by the addition of 250 μL of DI water. DI water was employed as the blank, gallic acid (GA) was used as the standard; the absorbance of all samples was measured at 750 nm, and results of total polyphenols were expressed as mg gallic acid equivalent per gram sample (GAE/g).

#### 2.3.2. Total Flavonoid Content

The total flavonoid content was determined as described by Eom et al. [15] according to the aluminum chloride method with some modification. Briefly, 250 μL of sample was mixed with 50 μL of aluminum nitrate (10%, *w*/*v*) and 50 μL of potassium acetate (1 M). The mixture was adjusted to 2 mL by the addition of 1650 μL of 80% methanol, vortexed, and allowed to stand for 40 min. Methanol (80%) was used as the blank control, with quercetin (QE) as the standard, and the absorbance of all samples was measured at 415 nm. All results were expressed as mg QE equivalent per gram of sample (QE/g).

#### 2.3.3. Total Tannin Content

The total tannin content (TT) was determined by a modification of the Folin–Ciocalteu method according to Makkar et al. [16] using polyvinylpolypyrrolidone (PVPP) to separate tannins from other phenols. Briefly, 1 mL of the Miang extract sample was mixed with 1 mL of 10% (*w*/*v*) PVPP, vortexed, and kept at 4 °C for 15 min. Then, the reaction mixture was centrifuged at 3000 rpm for 10 min, and the supernatant was collected. The remaining total phenol content of the PVPP precipitated supernatant was measured with the Folin–Ciocalteu reagent, and TT was estimated using the following formula: TT = TP − PVPP precipitated supernatant. The absorbance was measured at 750 nm for all samples. All results were expressed as mg tannic acid equivalent per gram of sample (TAE/g).

#### 2.3.4. Condensed Tannin Content

The contents of condensed tannin in Miang extracts were determined using the method described by Porter, et al. [17]. Briefly, butanol–HCl reagent (butanol–HCI 95:5 v/v) was prepared by mixing 950 mL of n–butanol with 50 mL concentrated HCl (37%). Ferric reagent (2% ferric ammonium sulfate in 2 N HCl) was prepared by dissolving 2.0 g of ferric ammonium sulfate in 2 N HCl (16.6 mL of concentrated HCl was made up to 100 mL with distilled water to make 2 N HCl). The reagents were stored in dark bottles. In a 100 mm × 12 mm glass test tube, 0.5 mL of Miang extract diluted with 70% acetone was pipetted. The quantity of acetone was large enough to prevent the absorbance (550 nm) in the assay from exceeding 0.6. Three milliliters of the butanol–HCl reagent and 0.1 mL of the ferric reagent were added to the tubes. The tubes capped with a glass marble were shaken using a Vortex and then placed on a heating block adjusted at 97 to 100 °C for 60 min. After cooling the tubes, absorbance was recorded at 550 nm. Absorbance of the unheated mixture (considered a suitable blank) was subtracted from the absorbance of the heated mixture, which was the actual reading at 550 nm to be used for calculation of condensed tannins. Development of pink color without heating the sample indicates the presence of flavan–4–ols. When this happened, one heated blank for each sample, comprising 0.5 mL of the extract, 3 mL of butanol, and 0.1 mL of the ferric reagent, was used. Condensed tannins (% in dry matter) as leucocyanidin equivalent were calculated by the formula:(A550 × 78.26 × Dilution factor)/(% dry matter)(1)

This formula assumes that the effective mass extinction coefficient for a 1% solution of leucocyanidin (El%), which absorbs at 550 nm in a 1 cm cuvette, is 460. Here, the dilution factor was equal to 1 if no 70% acetone was added and the extract was made from 200 mg sample in 10 mL solvent. Where 70% acetone was added (for example, to prevent the absorbance from exceeding 0.6), the dilution factor was 0.5 mL/(volume of extract taken).

#### 2.3.5. High Performance Liquid Chromatography (HPLC)

The extracted solutions were filtered with a 0.45 μm nylon membrane before injection into the HPLC system. The HPLC conditions were as follows: C18 column (250 × 4.6 mm, Phenomenex Gemini); UV detector at 210 and 254 nm; mobile phase (A) 0.1% acetic acid in acetonitrile and mobile phase (B) 0.1% acetic acid in DI water at a flow rate of 1.0 mL/min at 20 °C and injection volume 10 μL. The concentrations of catechins, organic acids, and caffeine were calculated based on their retention time (RT) compared with standards that included (−)-epigallocatechin gallate (EGCG), (−)-epicatechin gallate (ECG), (−)-gallocatechin gallate (GCG), (−)-epigallocatechin (EGC), (−)-epicatechin (EC), (+)-catechins (C), (−)-gallocatechin (GC), acetic, citric, gallic, galactcuronic, lactic, malic, succinic, tartaric, ellagic acids, and caffeine (Sigma–Aldrich, St. Louis, MO, USA).

### 2.4. In Vitro Antioxidant Assays

#### 2.4.1. DPPH Radical Scavenging Activity Assay

The 1,1-diphenyl-2-picrylhydrazyl (DPPH) radical scavenging activity of all samples was determined as described by Sánchez-Moreno et al. [18] with some modifications. DPPH solution at 0.15 mM was prepared in 80% methanol; 400 μL of the DPPH solution was mixed with 100 μL of different concentrations of individual samples. The sample mixture was allowed to incubate for 30 min in the dark, and the absorbance values were measured at 517 nm. Ascorbic acid was used as a positive control, 80% methanol was used as reagent blank, and the half-maximal inhibitory concentration (IC_50_) (μg/mL) was calculated for all samples from the plot of inhibition against extract concentrations following the equation:Inhibition (%) = (A_control_ − A_sample_/A_control_) × 100(2)
where A_control_ represents the absorbance of the mixture excluding the extract and A_sample_ is the absorbance of the DPPH solution containing the sample.

#### 2.4.2. ABTS Radical Scavenging Activity Assay

The ABTS radical scavenging activity was determined according to the method described by Radulović et al. [19] with slight modifications. Briefly, a 2,2′-azino-bis (3-ethylbenzothiazoline-6-sulfonic acid) radical cation (ABTS^+^) solution was prepared by mixing 7 mM solution of this sulfonic acid with 2.45 mM solution of potassium persulfate (K_2_S_2_O_8_) at 1:0.5 (*v/v*) followed by incubation for 12 h in the dark at 23 °C. The ABTS^+^ solution was diluted with 80% ethanol to attain an absorbance value of 0.700 ± 0.020 at 734 nm. A total of 20 μL aliquots of samples were mixed with 700 μL of the ABTS^+^ solution, and the mixtures were vortexed properly. Incubation was done for 6 min in the dark at 23 °C. The optical density (OD) was measured at 734 nm with 80% ethanol as the blank, and ascorbic acid served as a positive control.

#### 2.4.3. Ferric Reducing Antioxidant Power (FRAP) Assay

The ferric reducing antioxidant power of the extracts was determined as described by Benzie and Strain [20] with slight modification. Briefly, the FRAP working reagent consisted of 300 mM acetate buffer (pH 3.6), 10 mM 2,4,6-tri (2-pyridyl)-s-triazine (TPTZ) solution, and 20 mM ferric chloride at 10:1:1 (*v*/*v*/*v*). A total of 900 μL of working FRAP was mixed with 30 μL of sample (in triplicate). The mixture was allowed to incubate for 30 min, and absorbance was measured at 593 nm. FeSO_4_ was used as standard, and results were expressed as μmol Fe II equivalent per gram of extract.

### 2.5. Inhibition of Induced Intracellular ROS Production

The inhibition of intracellular ROS production induced by hydrogen peroxide (H_2_O_2_) was determined following the method of Saenjum et al. [21] and Phromnoi et al. [22] with slight modifications. The assay measures the oxidation of 2′,7′-dichlorodihydrofluorescein diacetate (H_2_DCF-DA) to fluorescent 2′,7′-dichlorofluorescein (DCF). Colorectal cells (HT-29) were seeded in a 96-well plate (5 × 10^3^ cells/well) and pre-treated with the test sample (10, 25, 50, and 100 µg/mL) or positive control and standards including 50 µM N-acetyl cysteine (NAC), catechin (125 µM) and 250 µM L-ascorbic acid for 12 h, followed by treatment with 500 μM H_2_O_2_ for 1 h to stimulate ROS production. Finally, 40 µM DCFH-DA solution was added, and fluorescence was detected by the excitation and emission wavelengths of 480 and 525 nm, respectively. All tested samples were treated in triplicate.

### 2.6. Inhibition of Pro-Inflammatory Nitric Oxide (NO) Production

The inhibition of induced nitric oxide production was determined using the improved method of Sirithunyalug et al. [23] with some modifications. Initially, RAW 264.7 cells were pre-incubated in a 96-well plate (5 × 10^3^ cells/well) for 24 h. Then, cells were replaced with a fresh medium containing various concentrations of test samples at a concentration of 10, 25, 50, and 100 μg/mL. After 12 h of incubation, combined lipopolysaccharide (LPS) and interferon-gamma (IFN-γ) were added. After 48 h incubation in 5% CO_2_ at 37 °C, the culture medium supernatants were collected and analyzed for nitric oxide. The quantity of nitrite using Griess reagent in the culture medium was measured at 540 nm as an indicator of nitric oxide production; fresh culture medium was used as a blank, and curcumin and catechin were used as the positive control.

### 2.7. Statistical Analysis

All experiments were carried out in triplicate. The data are expressed as means ± SD and were evaluated using analysis of variance (ANOVA). Statistical analyses were carried out using the Statistical Package for the Social Sciences statistical software (Version 20, SPSS Inc., Chicago, IL, USA). *p* < 0.05 was considered statistically significant.

## 3. Results and Discussion

### 3.1. Chemical Composition of Miang

Wide sources of research publications have attested to the suitability of acetone as an extraction solvent for plant phenolics [24,25]. A total of 80% acetone was used in this study to extract young tea leaves (YTL), mature tea leaves (MTL), NFP-Miang, and FFP-Miang. The crude extract from all samples was dissolved in DI water and used for phytochemical determinations of the total polyphenol (TP), total tannin (TT), condensed tannin (CT), and total flavonoid (TF) contents (Figure 2).

The contents of TP, TT, and CT increased significantly (*p* < 0.05) after fermentation (NFP-Miang: 558.51 mg/g, 347.27 mg/g, 198.36 mg/g and FFP-Miang: 285.25 mg/g, 217.27 mg/g and 170.04 mg/g, respectively) compared to their respective starting materials, viz. young leaves (TP: 242.49 mg/g; TT: 167.31 mg/g and CT: 43.12 mg/g) and mature leaves (TP: 206.53 mg/g; TT: 120.38 mg/g and CT: 27.01 mg/g). However, the TF content was not significantly (*p* > 0.05) different for NFP-Miang (22.16 mg/g) compared to young leaves (20.26 mg/g). On the contrary, fermentation seemed to significantly lower the TF contents in FFP-Miang (4.69 mg/g) compared to mature tea leaves (24.06 mg/g). These data suggest that exclusive bacterial and yeast fermentation (as seen in NFP) as well as stepwise filamentous fungal and bacterial fermentation (FFP) of tea leaves play a major role in the elevated levels of these phenolic secondary metabolites. The increase in TP (~2-fold and 1.38-fold for NFP-Miang and FFP-Miang) and TT (~2-fold and 1.80-fold, respectively) could be attributed to microbial hydrolytic enzymes (such as esterases and glucosidases) capable of lysing complex phenolics, thereby releasing free absorbable phenolic compounds that are easily detected [26]. Other reasons for this increase might include an increase in the extractability of tea polyphenols and microbial biosynthetic and/or hydrolytic enzymes (such as amylases, proteases, and xylanases) capable of causing structural modification of tea leaf cell walls during fermentation, thus mobilizing free polyphenols from complexed forms or synthesizing new bioactive phenolics from the available substrates [27,28,29]. The contents of CT also significantly increased (*p* < 0.05) after fermentation (NFP-Miang: 198.36 mg/g and FFP-Miang: 170.04 mg/g) compared to their respective starting materials (young leaves: 43.12 mg/g and mature leaves: 27.01 mg/g). This ~5-fold and 6-fold (NFP-Miang and FFP-Miang) increase in CT could be explained in light of the possible oxidation of monomeric flavan-3-ols by microbial enzymes [30]. The decrease in TFs seen in FFP-Miang (Figure 2) corroborates this reason for the increase in CT post fermentation. Alternate mechanisms might be involved in the synthesis of condensed tannin during microbial fermentation considering the levels of TFs in NFP were not significantly altered after fermentation. Additionally, fractionated Oolong tea extracts have been shown to contain higher amounts of TPs, TFs, and CTs compared to the original Oolong tea infusion [31]. The contents of catechins and some other phenolic compounds in Miang samples were determined by HPLC although the result showed a general decline in the contents of detected catechins (Table 1).

The concentrations of gallic acid increased from 0.57 mg/g (young leaves) and 0.26 mg/g (mature leaves) to 1.06 mg/g (NFP-Miang) and 0.68 mg/g (FFP-Miang), which corresponded to a ~2- and 3-fold increase, respectively, after fermentation. The gallic acid could have emanated from bacterial or fungal hydrolysis of monomeric or polymeric galloylated phenolics such as other catechins and their flavan-3-ol polymers. The contents of gallic acid could also have been augmented by microbial enzymes with specificity towards ester and ether bonds that link sugars to phenolic acids and adjoining compounds in hydrolyzable tannins (gallotannins) [32,33]. Weng, et al. [34] reported a 44.8-fold increase in the gallic acid content when Oolong tea was post fermented with *Aspergillus sojae*. Depending on the type of fermented tea, the contents of gallic acid can be as high as 21.98 mg/g as shown for Pu-erh tea [32]. Mature tea leaves had the highest catechin content (21.82 mg/g). The catechin content was not significantly different (*p* > 0.05) in young leaves (6.97 mg/g) and NFP-Miang (6.89 mg/g), while FFP-Miang showed the lowest detectable catechin (4.75 mg/g). Epicatechin (EC) was only detected in young leaves (2.07 mg/g) and mature leaves (1.75 mg/g), not in the fermented samples. Although the ECs were low, it could be theorized that catechin and EC may have been substrates for microbial oxidases in the fermented samples, which in turn catalyzed oxidative polymerization to produce procyanidin (a class of cis-cis proanthocyanidins produced by the oxidation of catechin and EC linked through C4-C8 or C4-C6 covalent bonds) [35,36,37]. The contents of epigallocatechin-gallate (EGCG), epicatechin gallate (ECG), and caffeine followed the aforementioned decline. Pyrogallol went undetected in the unfermented tea leaves but was found at relatively low levels in NFP-Miang and FFP-Miang (0.21 mg/g DW and 0.26 mg/g DW, respectively). The pyrogallol-type catechins such as EGCG and EGC are already established as major bioactive catechins of tea [38]. Therefore, galloylation of phenolics in the microbial fermented tea (NFP-Miang and FFP-Miang) might be suggested as a possible reason for their improved bioactivities. The contents of organic acids were also determined by HPLC and the result showed an interesting overall increase in organic acid levels post-fermentation (Figure 3).

Ellagic acid was detected in Miang extracts at concentrations of 1.90 mg/g (NFP-Miang) and 1.13 (FFP-Miang), which suggests that hydrolyzable tannins (ellagitannins) in the fermented samples might be the sources of this bioactive organic acid. In addition, ellagic acid has been reported in Fuzhuan tea, which is produced via fermentation with the yellow fungus *Aspergillus cristatum*. Pyrogallol, gallic acid, and ellagic acid appear to be important pharmacophores that confer fermented Miang extracts with requisite bioactivities as their appearance increased in the NFP- and FFP-Miang samples, which suggests that is the case (Figure 4).

These structures seem to also translate into bioactivities of condensed tannins as their B-rings have been shown to be closely similar to ones present in parent phenolics [39]. The catechol- and/or pyrogallol-related B-rings of condensed tannins are reported to mediate reactions such as electrophilic additions, metal chelation, scavenging of free radicals, and oxidation required to potentiate the bioactivities of these compounds [39]. NFP-Miang and FFP-Miang contained 57.12% and 78.26% of CT, which may have arose from enzymatic and/or oxidative coupling of pyrogallol rings (EGCG-EGCG coupling) or via condensation of EC (catechol-type B-ring) to pyrogallol type rings such as EGCG and EGC [40]. The pyrogallol-type B-rings are easily susceptible to oxidation because of their low redox potential, thus allowing CT-increased hydroxyl (OH) groups and electron delocalization sites that play major roles in their bioactivities [41]. These bioactive compounds have always shown bioactivity in vitro but mildly in vivo in many cases due to low bioavailability in tissues [42]. They are more bioavailable in plant extracts compared to when administered in their pure forms to laboratory animals as has been reported for free ellagic acid (EA) and EA in pomegranate extract [43]. Microbial cellulases, polyphenol oxidases, β-glucosidases, tannases, and xylanases have been implicated in the biodegradation of ellagitannins that result in the generation of ellagic acids [43,44]. The mechanism of EA release from ellagitannins is yet to be fully elucidated albeit EA has been found to be converted to a more absorbable dibenzopyranone metabolite, viz. urolithins in the gastrointestinal tract (GIT) [45]. Urolithins have been reported to possess anti-inflammatory effects as well as anticancer effects on colon and prostate cancer cells [46,47,48]. Contents of galacturonic acids increased also after fermentation from 9.59 mg/g (mature leaves) to 89.99 mg/g in FFP-Miang. There was no detectable concentration of galacturonic acid in young leaves but its content in NFP-Miang was 79.03 mg/g. The galacturonic acid is thought to increase due to microbial degradation of Miang pectins by pectin esterases or some other hydrolytic enzymes in the course of fermentation [49]. Some bacteria have been identified to be capable of synthesizing tartaric acids from a succinic acid derivative (cis-epoxysuccinate) and 5-ketogluconic acid. Thus, the increase in tartaric acid from 3.67 mg/g (young leaves) and 3.87 mg/g (mature leaves) to 6.34 mg/g (NFP-Miang) and 4.95 mg/g (FFP-Miang) observed post fermentation in Miang could have occurred through a similar or related pathway. Succinic acid is a metabolite of the Krebs cycle, which also increased post-fermentation from 7.30 mg/g (young leaves) and 9.77 mg/g (mature leaves) to 24.82 mg/g (NFP-Miang) and 25.27 mg/g (FFP-Miang). Fungal cultures in FFP evidently contribute to the succinic acid contents in FFP-Miang due to their aerobic catabolism, which allows them to use sugars in tea as substrates. Some facultative anaerobes such as *Enterococcus* spp. and *Lactobacillus* species have been reported to produce succinic acids using glucose as a carbon source albeit a different study showed that fumaric acid with glycerol serving as hydrogen donor also led to successful production of succinic acid by these microbes [50,51]. This helps explain the succinic acid levels observed in NFP-Miang given that the fermentation is carried out under anaerobic conditions by the aid of bacteria. Acetic acid was only detected in FFP-Miang at a concentration of 11.35 mg/g. This is no surprise as acetic acid could be synthesized from acetyl-CoA (which might originate from fungal aerobic metabolism of sugars in FFP-Miang) Thus, these organic acids, alone or in synergy, contribute to the elevated bioactivities of fermented teas. The increase in lactic acid after fermentation suggests activities of anaerobic lactic acid-producing bacteria in both NFP-Miang and FFP-Miang. This organic acid increased from 4.00 mg/g (young leaves) and 7.41 mg/g (mature leaves) to 59.52 mg/g (NFP-Miang) and 43.67 mg/g (FFP-Miang), respectively.

### 3.2. Antioxidant Activities of Miang Extracts

The antioxidant activities of Miang extracts have been previously reported in the literature [9]. In this study, DPPH, ABTS, and FRAP assays were used to investigate the free radical scavenging activities of Miang extracts. The DPPH radical accepts an electron or is reduced by hydrogen, as is the less stable ABTS radical, to become a stable diamagnetic molecule [52]. Given the similarity between the DPPH and ABTS assay principles, results from both assays corroborate antioxidant functionality that might not be ideal in all cases [53]. The FRAP assay evaluates the ability of plant phenolics to reduce ferric ion to its corresponding ferrous form under acidic conditions (pH 3.6). This is usually achieved via electron donation to ferric ion by phenolic compounds. Fermentation improved the DPPH radical scavenging activities of NFP-Miang and FFP-Miang, which showed IC_50_ values of 13.15 μg/mL and 9.31 μg/mL, respectively (Table 2). These IC_50_ values were not significantly different (*p* > 0.05) from that recorded for the control ascorbic acid (7.39 μg/mL). Similarly, the ABTS radical scavenging activity of NFP (18.50 μg/mL) and FFP (13.87 μg/mL) was not significantly different (*p* > 0.05) from the ascorbic acid control (8.85 μg/mL) (Table 2). Thus, in both DPPH and ABTS assay results, fermented tea extracts (NFP-Miang and FFP-Miang) had better IC_50_ values compared to young and mature leaf extracts: NFP-Miang = FFP-Miang = ascorbic acid < 34.52 μg/mL (young leaves) < 37.76 μg/mL (mature leaves) for the DPPH assay and NFP-Miang = FFP-Miang = ascorbic acid < 36.98 μg/mL (young tea leaves) < 42.17 μg/mL (mature tea leaves) for the ABTS assays.

The nonsignificant difference observed for NFP-Miang and FFP-Miang in relation to ascorbic acid is interesting, as the ability of this control to scavenge superoxide ion, hydroxyl radical anion, reactive nitrogen oxide, hydrogen peroxide (H_2_O_2_), and singlet oxygen has been confirmed in the literature [54]. DPPH is an organic compound that dissolves in solvents such as ethanol/methanol, thus making DPPH assay an estimator of hydrophobic antioxidants, which is contrary to the lipophilic and/or hydrophilic ABTS^+^ radical, which possibly accounts for both hydrophobic and hydrophilic phenolics in Miang extract [55]. The phenolic antioxidant activities rely largely on the structure of polyphenols and number and position of OH groups, as OH groups that occupy the ortho position of pyrogallol B-ring (Figure 4) have been implied to contribute to the antioxidant capabilities of phenolic compounds [56]. Oxidized phenolics appear to give an advantage to NFP- and FFP-Miang extracts considering the increase in condensed tannin, pyrogallol, and gallic acid observed in Figure 2 and Table 1, which coincided with better DPPH and ABTS radical scavenging activities (Table 2). Gallic acid and pyrogallol have been shown to possess the strongest H_2_O_2_ and DPPH radical scavenging activities compared to other hydrophilic phenolics [57]. Additionally, a molecule of GA was able to reduce six DPPH radicals, thus showing a high potential for radical scavenging [58]. The extracts also showed FRAP capability where the concentrations of Fe^2+^ increased after the experiment, suggesting electron transfer between phenolics and available Fe^3+^: NFP-Miang > FFP-Miang = young leaves > mature leaves > Iron (II) sulphate (9266.47 μmol Fe^2+^/g > 7236.44 μmol Fe^2+^/g = 7880.49 μmol Fe^2+^/g > 5004.69 μmol Fe^2+^/g > 3597.122 μmol Fe^2+^/g) (Table 2). It could be suggested that fermentation and the fermenting organisms directly or indirectly contribute to the observed antioxidant activities. These activities are usually achieved via electron or hydrogen transfer from polyhydroxylated monomeric and/or condensed phenolics in tea onto these free radicals and trace metals (Fe^3+^), thus acting as reducing agents, quenchers of singlet oxygen, and radical scavengers [59,60]. Bacterial association with plants allows for easy absorption of Fe^3+^ due to the ability of siderophores secreted by these microbes alongside phytosiderophores to form soluble complexes with ferric ion [61], which implies that an additional source of the much needed ferrous ion (Fe^2+^) increases as a result of tea fermentation, which could be credited to the phenolic groups being actively responsible as previously investigated. Metal chelation by phenolic compounds is quite important as they facilitate a decrease in transition metal concentrations, thus preventing their role in the catalysis of lipid peroxidation [62]. Giving the propensity for gallated catechins to be oxidized in NFP- and FFP-Miang, it could be suggested that the pyrogallol and gallic acid moieties present in both monomeric and oxidized phenolics provided phenolics in fermented Miang samples with a high susceptibility for nucleophilic attack, thereby contributing to the reduction of transition metals (Fe^3+^) as well as radical quenching. This is possible since gallated catechins have been found to possess lower energy gaps between their highest occupied molecular orbital (HOMO) and lowest unoccupied molecular orbital (LUMO) compared to non-galloylated phenolics [63]. The high contents of condensed tannins in both NFP- and FFP-Miang could also have decreased the bioavailability of metals ions such as iron, zinc, and calcium ions via the formation of insoluble complexes as has been reported for gallic acid and tannic acid [64,65,66]. Lactic acid as well as butyric acid found in Miang extracts has been suggested to hold much potential in therapeutics [6,9]. The lactate ion is strongly suggested to mitigate/prevent lipid peroxidation by scavenging superoxide and hydroxyl radicals in vitro [67]. Butyric acid was similarly reported to mitigate oxidative stress in streptozotocin-induced diabetic rats by improving the activities of catalase, superoxide dismutase, and glutathione peroxidase [68]. Gallic acid is a low-molecular-weight triphenolic acid found in almost all traditional fermented teas and is renowned for its strong antioxidant, antimicrobial, anti-inflammatory, and anticancer activities [32]. It is reported to neutralize reactive species via metal chelation, radical scavenging, apoptosis induction, cell signaling pathway interference, and inhibition of lipid peroxidation [69]. Similarly, ellagic acid is thought to neutralize the effects of oxidative stress by acting as an antioxidant directly or indirectly through induction of antioxidant systems in the cell to counter the stress [70]. Other organic acids such as acetic acid, glucuronic acid, galacturonic acid, caffeic acid, gluconic acid, chlorogenic acid, malic acid, oxalic acid, citric acid, and derivatives of benzoic acid found in these traditional fermented teas play typical roles as intermediaries in metabolism and directly or indirectly impact the antioxidant potentials of teas [6,71,72].

### 3.3. Inhibition of the Intracellular Production of Reactive Oxygen Species (ROS) in a Cell-Based Study

The protective effects of Miang extracts against H_2_O_2_-induced ROS production was investigated in HT-29 colorectal cells. The cells were pre-incubated with different concentrations (10, 25, 50, and 100 ppm) of extracts and/or positive controls, viz. catechin (125 µM), L-ascorbic acid (250 µM), and 50 µM of N-acetylcysteine (NAC). The results showed that NFP-Miang, FFP-Miang, and mature tea leaves showed significant inhibition of ROS production at 50 ppm and 100 ppm concentrations as the ROS levels were not significantly different (*p* > 0.05) from the positive controls used. The percentage of ROS recorded for cells preincubated with NFP-Miang, FFP-Miang, and mature leaves at a concentration of 50 ppm was 68.05%, 57.54%, and 77.10%, while 51.45%, 42.70%, and 61.69% was recorded for 100 ppm extract concentrations of NFP-Miang, FFP-Miang, and mature leaves, respectively, compared to positive controls: catechin (60.04%), L-ascorbic acid (67.55%), and NAC (51.89%). A 100 ppm concentration of young leaf extracts (68.96%) was able to inhibit ROS production in a similar range to the positive controls as shown in Figure 5.

This suggests that fermented samples have much higher protective capacities towards inhibiting ROS accumulation in these cells, as lower extract concentrations (50 ppm) could cause an approximate 50% decrease in ROS generation compared to positive controls. The amount of condensed tannins in fermented Miang might be the reason for the improved inhibition, as the radical scavenging ability of tea phenolics generally depends on the number of hydroxyl (OH) groups surrounding the structures, the type of environment, and how stable phenolic oxygen radicals are after ROS scavenging [73]. These phenolics might as well act indirectly through antioxidant enzymes such as glutathione peroxidase (GPx) and catalase capable of neutralizing the harmful effect of H_2_O_2_ in cells [74,75]. Polyphenols have also been shown to inhibit ROS production at a genetic level as they promote the translocation of nuclear factor erythroid-derived 2-like 2 (Nrf2) from the cytosol to the nucleus where it binds the antioxidant response elements (AREs) to induce the expression of important antioxidant enzymes such as catalase (CAT), superoxide dismutase (SOD), heme oxygenase-1 (HO-1), and quinone oxidoreductase-1 (NQO1) [76,77]. Cascades of other metabolic and signaling pathways are targets for tea polyphenols, thus the antioxidant activities can be potentiated through various mechanisms [78]. The presence of oxidized phenolics in NFP- and FFP-Miang in consonance with the previously mentioned pharmacophores might be responsible for their improved reactive oxygen species quenching since theaflavins usually found in fermented teas have a higher reaction rate than EGCG in the scavenging of superoxide radicals [63]. A GA level of 5 μM was able to decrease levels of ROS in phorbol ester-activated macrophages. In addition, molecular docking studies showed that the carboxyl group (COOH) of GA mediated a stronger interaction between GA and PKCδ (a receptor targeted by tumor promoters such as phorbol ester), thus inhibiting PKCδ activation in silico [32]. Gallic acid increased levels of glutathione peroxidase (GPx), superoxide dismutase (SOD), catalase (CAT), glutathione reductase (GRx), and glutathione S-transferase (GST) when pretreated rats were injected with isoproterenol [79]. The gallic acid dimer of ellagic acid has been shown to inhibit leukocyte accession to the endothelium via ROS inhibition [80]. Other microbial fermented teas such as Kombucha have been shown to protect murine hepatocytes against tertiary butyl hydroperoxide (TBHP)-induced ROS production via inhibition of the mitochondrial dependent pathway (prevented changes in the mitochondrial membrane potential, cytochrome *c* release, as well as inhibited the activation of caspases (3 and 9) and Apaf-1) [81]. Lactic acid bacteria (LAB)-fermented black tea showed a better inhibition of H_2_O_2_-induced ROS production in human colonocytes compared to conventional black tea where the fermented black tea repaired DNA and mitigated lesions and breaks of DNA strands [82].

### 3.4. Inhibition of Nitric Oxide (NO) Production

The inhibitory effect of Miang extracts against LPS- and IFN-γ-induced nitric oxide production in RAW 264.7 cells was investigated. The result showed better IC_50_ values for fermented teas compared the young and mature tea leaves (curcumin < catechin = NFP-Miang < FFP-Miang < young leaves = mature leaves); curcumin and catechin were used as positive controls (Figure 6). The IC_50_ values were 6.85 μg/mL (curcumin), 15.49 μg/mL (catechin), 17.15 μg/mL (NFP-Miang), 20.17 μg/mL (FFP-Miang), 33.96 μg/mL (young leaves), 31.33 μg/mL (mature leaves).

It is interesting to note once again that fermentation contributes immensely to the increased bioactivity of these tea extracts. This inhibition might be mediated via a number of pathways including scavenging of NO radicals and its metabolites such as nitrite and nitrate radicals and/or scavenging of the highly reactive peroxynitrites, which are formed due to superoxide radicals reacting with NO radicals [83,84]. NO is a proinflammatory mediator, thus the lower IC_50_ values for fermented teas compared to unfermented leaves could be due to the inhibition of inducible nitric oxide synthase (iNOS) enzyme or by reducing iNOS gene expression [31]. Black teas, Oolong tea, and microbial fermented teas such as Kombucha and Pu-erh have also been reported to inhibit inflammation and related disease pathways, viz. atherosclerosis and cancer models via inhibition of NO production through iNOS, as well as inhibit other proinflammatory mediators such as interleukin-12, monocyte chemoattractant protein (MCP), and tumor necrosis factor-α (TNF-α) [31,40,85]. The result from this study corroborate positive impacts that have been reported for other traditional fermented teas such as Pu-erh, Oolong, Zijuan, and Kombucha, among others, considering the fermentation of both Miang and these teas’ is mediated by microbes [6,86]. Flavonoid aglycones such as kaempferol, quercetin, myricetin, apigenin, and their glycosides (flavonoid glycosides) detected in fermented teas such as Tuocha (Chinese bowl tea), Pu-erh, and Oolong tea are suggested to contribute to the bioactivities of these teas [87]. It is worth noting that although the flavonoid aglycones and their glycosides are not yet reported in fermented tea such as Miang, the related raw materials (tea leaves) and fermentation process support that Miang especially holds so much promise as a prospective repository of these compounds and perhaps other novel compounds as well considering the different microorganisms involved in the fermentation of different Miang products as well as the diversity of its fermentation processes [6]. Large molecular polymeric pigments (LMPP) found in Zijuan tea appear to confer significant bioactivity to this fermented product, as antioxidant and antiobesity effects of this tea have been partly attributed to LMPP [11,88]. This could be correlated with the suggested bioactivity in NFP-Miang and FFP-Miang, which is partly contributed to by condensed tannins (large polymeric flavonoids). In addition, conjugated tannins (molecular wt. range of 1000–10,000 daltons) in Miang extract have been presumed as the factors responsible for the inhibitory effect of DNA synthesis in cultured rat mammary tumor cells [13]. As shown in Figure 2, Figure 3 and Figure 5 and Table 1 and Table 2, fermentation appeared to increase the concentrations of bioactive compounds (gallic acid, ellagic acid, pyrogallol, and condensed tannin) as well as conferred improved antioxidant and anti-inflammatory activities on NFP- and FFP-Miang extracts compared to young and mature tea leaves. This observation is also true for some traditional fermented teas such as Pu-erh and Kombucha. Fermentation has been reported to increase the contents of gallic acid, kaempferol, quercetin, resorcylic acid, and aglycones in Pu-erh compared to unfermented green tea, thus conferring a better anticancer and antiangiogenesis effect to Puerh tea (with an 85% inhibition in fermented tea compared to unfermented Puerh leaves and green tea, which showed 67% and 53% inhibition, respectively).

These research findings offer a peek into the biological activities of traditional fermented Miang extracts (NFP- and FFP-Miang). The door is still open for further investigations, especially in the case of Miang where only one promising research article has been published regarding its anticancer effects [13].

## 4. Conclusions

Microbial biotransformation of phenolics contributes to improved enrichment of total polyphenols, total tannins, and condensed tannins in NFP-Miang and FFP-Miang compared to unfermented tea leaves. The overall phenolics increase correlated with higher radical scavenging activity, inhibition of intracellular reactive oxygen species/reactive nitrogen species, as well as inhibition of a proinflammatory mediator (NO). Pyrogallol, gallic acid, and ellagic acids may be the pharmacophores that confer the improved bioactivities to NFP- and FFP-Miang. Thus, these fermented tea leaves of Miang could find relevant applications as bioactive functional ingredient targets for countering inflammatory-mediated diseases such as cancer and cardiovascular diseases. This provides the foundation to study the phenolics responsible for these bioactivities of NFP-Miang and FFP-Miang in in vivo and animal models.

## Figures and Tables

**Figure 1 antioxidants-10-01144-f001:**
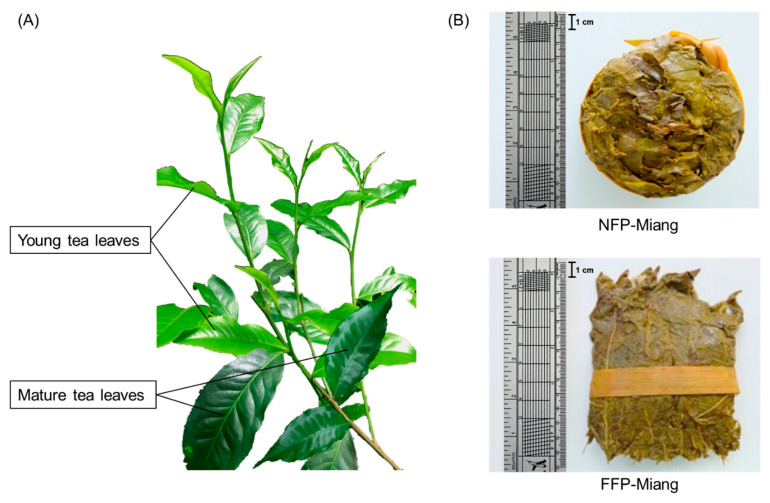
The appearance of tea leaves, i.e., young and mature leaves (**A**), non-filamentous fungi-process-based fermented leaves (NFP-Miang) and filamentous fungi-process-based fermented leaves (FFP-Miang) (**B**).

**Figure 2 antioxidants-10-01144-f002:**
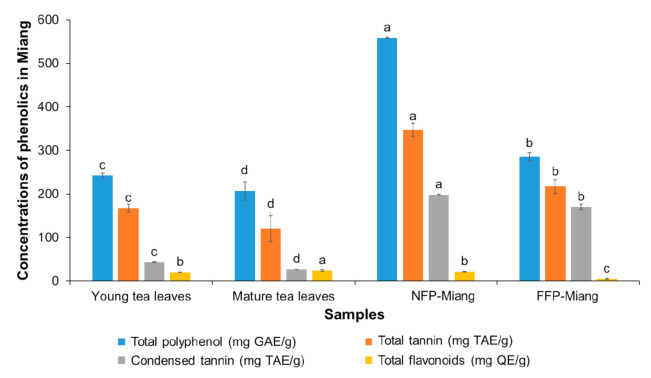
The contents of phenolics in Miang samples: Young tea leaves, mature tea leaves, non-filamentous fungi-process-based fermented leaves (NFP-Miang), and filamentous fungi-process-based fermented leaves (FFP-Miang). Different letters represent significant differences in group concentrations at *p* < 0.05. Gallic acid equivalent (GAE), tannic acid equivalent (TAE), quercetin equivalent (QE).

**Figure 3 antioxidants-10-01144-f003:**
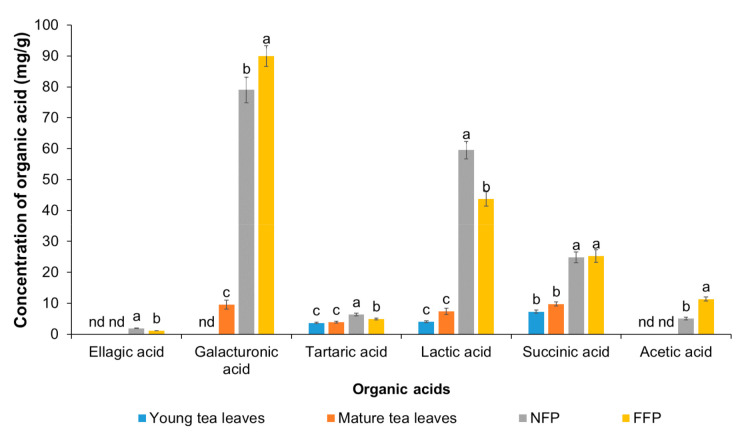
The contents of organic acids in Miang samples: Young tea leaves, mature tea leaves, non-filamentous fungi-process-based fermented leaves (NFP-Miang), and filamentous fungi-process-based fermented leaves (FFP-Miang). Different letters represent significant differences in group concentrations at *p* < 0.05, and “nd” represents “not determined”.

**Figure 4 antioxidants-10-01144-f004:**
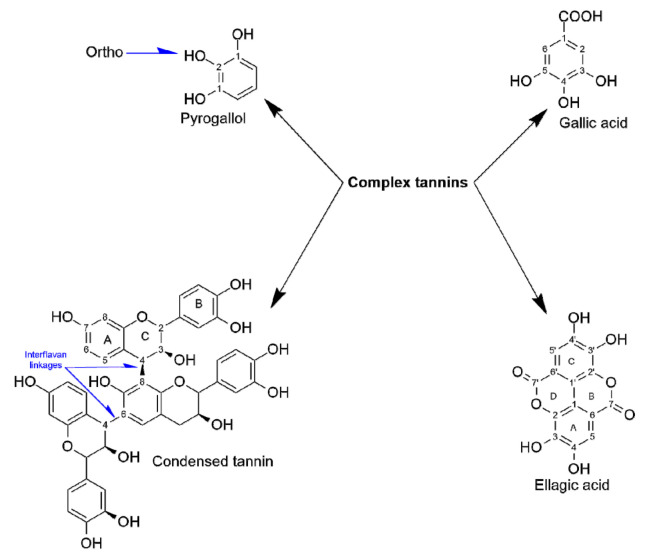
Phenolic structures and salient functional groups that contribute to NFP- and FFP-Miangs’ biological effects.

**Figure 5 antioxidants-10-01144-f005:**
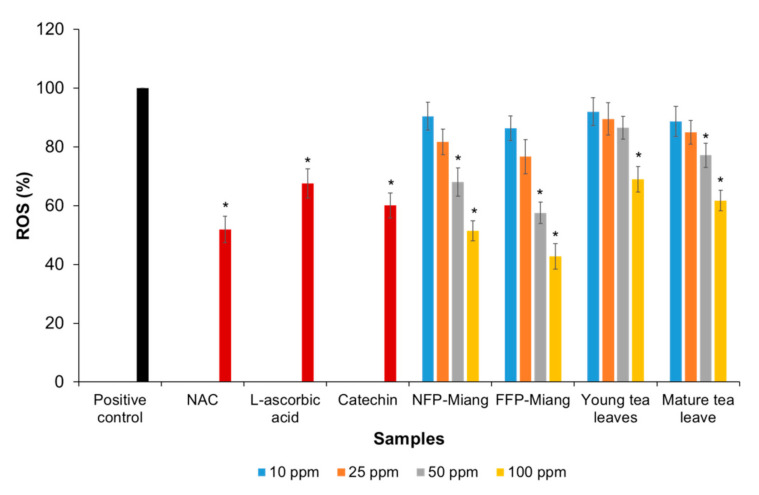
Percentage inhibition of various Miang extract concentrations toward intracellular reactive oxygen species (ROS) production in HT-29 colorectal cells induced by hydrogen peroxide using N-acetylcysteine (NAC, 50 µM), L-ascorbic acid (250 µM), and catechin (125 µM) as positive controls. Different asterisks (*) represent significant differences in group percentages at *p* < 0.05 vs. the positive control.

**Figure 6 antioxidants-10-01144-f006:**
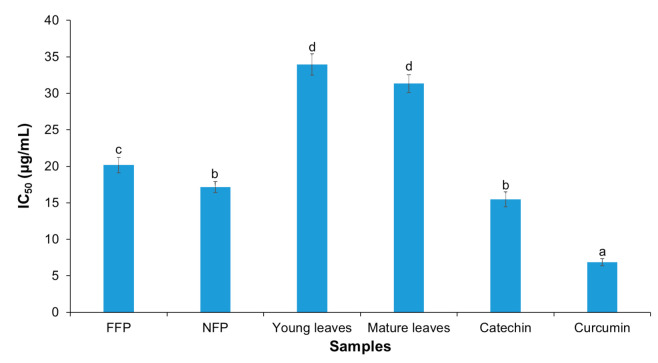
Inhibition of nitric oxide (NO) production by young tea leaves (NFP-Miang) and mature tea leaves (FFP-Miang). Catechin and curcumin were used as positive controls. Different letters represent significant differences in group concentrations at *p* < 0.05 vs. controls.

**Table 1 antioxidants-10-01144-t001:** The contents of catechin-related compounds and some bioactive compounds in Miang samples: Young tea leaves, mature tea leaves, non-filamentous fungi-process-based fermented tea leaves (NFP-Miang), and filamentous fungi-process-based fermented tea leaves (FFP-Miang).

Sample	Bioactive Compounds (mg/g Sample)						
	Catechin (C)	Epicatechin (EC)	Epigallocatechin Gallate (EGCG)	Epicatechin Gallate (ECG)	Pyrogallol	Gallic Acid (GA)	Caffeine
Young leaves	6.97 ± 0.14 ^b^	2.07 ± 0.08 ^a^	0.04 ± 0.003 ^d^	1.13 ± 0.06 ^b^	nd	0.57 ± 0.07 ^b^	1.04 ± 0.08 ^c^
Mature leaves	21.82 ± 1.13 ^a^	1.75 ± 0.06 ^b^	0.05 ± 0.002 ^c^	2.03 ± 0.07 ^a^	nd	0.26 ± 0.03 ^c^	1.98 ± 0.07 ^a^
NFP-Miang	6.89 ± 0.11 ^b^	nd	0.07 ± 0.003 ^a^	1.15 ± 0.07 ^b^	0.21 ± 0.007 ^b^	1.06 ± 0.02 ^a^	1.23 ± 0.08 ^b^
FFP-Miang	4.75 ± 0.13 ^c^	nd	0.06 ± 0.003 ^b^	nd	0.26 ± 0.005 ^a^	0.68 ± 0.03 ^b^	0.73 ± 0.05 ^d^

Different letters represent significant differences in group concentrations at *p* < 0.05, and “nd” represents “not determined”.

**Table 2 antioxidants-10-01144-t002:** Antioxidant activities of Miang samples determined by DPPH, ABTS, and FRAP assays using L-ascorbic acid as a positive control in comparison to FRAP assay using iron (II) sulfate as a standard.

Sample	DPPH IC_50_ (μg/mL)	ABTS IC_50_ (μg/mL)	FRAP (μmol Fe II/g)
Young leaves	34.52 ± 4.57 ^b^	36.98 ± 5.10 ^b^	7880.49 ± 561.82 ^b^
Mature leaves	37.76 ± 7.42 ^c^	42.10 ± 6.74 ^c^	5004.69 ± 358.92 ^c^
NFP-Miang	13.15 ± 2.26 ^a^	18.50 ± 2.88 ^a^	9266.47 ± 707.67 ^a^
FFP-Miang	9.31 ± 2.92 ^a^	13.87 ± 6.38 ^a^	7236.44 ± 788.23 ^b^
Ascorbic acid	7.39 ± 0.67 ^a^	8.85 ± 1.96 ^a^	-
Iron (II) sulfate	-	-	3597.12 ± 0.00 ^d^

Different letters represent significant differences in group concentrations at *p* < 0.05.

## Data Availability

The data is contained within the article.

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
