# Peer review of "Comparison of Phenolic Contents and Scavenging Activities of Miang Extracts Derived from Filamentous and Non-Filamentous Fungi-Based Fermentation Processes"

_antioxidants, 2021, doi:10.3390/antiox10071144_

Round 1
Reviewer 1 Report
Abstract: page 1, line 25-26: The sentence “ The antioxidant studies also showed lower IC50 values for fermented leaves in both DPPH and ABTS radical scavenging activity assays as well showed ferric reducing antioxidant power (FRAP) assay.” is not clear. Please rephrase the sentence.
Materials and Methods: page 3, line 107: Please check the word “tannin” in subtitle Total polyphenol content.
Materials and Methods: page 3, line 109: The abbreviation “DI” appears for the first time in the text, please define abbreviation in parentheses.
Materials and Methods: page 4, line 155: The sentence “This formula assumes that the effective El%, 1 cm, 550 nm of leucocyanidin is 460.” is not clear. Please rephrase the sentence. The abbreviation EI% appears for the first time in the text, please define abbreviation in parentheses.
Materials and Methods: page 4, line 166: The abbreviation “RT” appears for the first time in the text, please define abbreviation in parentheses.
Materials and Methods: page 5, line 193: The abbreviation “OD” appears for the first time in the text, please define abbreviation in parentheses.
Materials and Methods: page 5, line 202: Please check the word “μmole” in subtitle Ferric reducing antioxidant power (FRAP) assay.
Materials and Methods: page 6, line 220-221: The abbreviations “LPS” and “IFN-g” appears for the first time in the text, please define abbreviation in parentheses.
Results and Discussion: page 8, line 286: The reference in the text “ [34]” is cited wrong please corrected reference in “Weng et al. [34]”.
Results and Discussion: page 8, line 287: Check the words “Aspergillus sojae” and write in italics if necessary.
Results and Discussion: page 8-9, line 312, 335: Check the word “Ellagitannins” and write with small capital letter if necessary.
Results and Discussion: page 8-19, line 320, 324, 325, 326, 398, 400,402…: Check the word “Pyrogallol” and write with small capital letter if necessary.
Results and Discussion: page 9, line 329: Check the words “in vitro” and “in vivo” and write in italics if necessary.
Results and Discussion: page 9, line 334: The abbreviation “ETS” appears for the first time in the text, please define abbreviation in parentheses.
Results and Discussion: page 9, line 337: Check the word “Urolithins” and write with small capital letter if necessary.
Results and Discussion: page 10, line 396: Reference in the text “(Ozturk Sarikaya, 2015)” is incorrectly cited and is missing from the Reference list. Please cite the reference correctly in the Results and Discussion and in the Reference list.
Results and Discussion: page 13, line 501: The abbreviation “LAB-” appears for the first time in the text, please define abbreviation in parentheses.
Author Response
Response to Reviewer 1 Comments
We have revised the manuscript following all suggested points of reviewer and all changed points are highlighted with yellow color in R1 version. Regarding the question from reviewers, the answers or explanations are responded to one by one as follows;
Point 1: Abstract: page 1, line 25-26: The sentence “ The antioxidant studies also showed lower IC50 values for fermented leaves in both DPPH and ABTS radical scavenging activity assays as well showed ferric reducing antioxidant power (FRAP) assay.” is not clear. Please rephrase the sentence.
Response 1: The sentence was revised to ‘The antioxidant studies also showed better IC50 values for fermented leaves in both DPPH and ABTS radical scavenging activity assays as well showed improved ferric reducing antioxidant power (FRAP) when compared to young and mature tea leaves’ (line 26-28).
Point 2: Materials and Methods: page 3, line 107: Please check the word “tannin” in subtitle Total polyphenol content.
Response 2: It was a typographical error and has been changed to ‘polyphenol’ as suggested (line 110).
Point 3: Materials and Methods: page 3, line 109: The abbreviation “DI” appears for the first time in the text, please define abbreviation in parentheses.
Response 3: We have inserted ‘deionised (DI) water’ as suggested (line 112).
Point 4: Materials and Methods: page 4, line 155: The sentence “This formula assumes that the effective El%, 1 cm, 550 nm of leucocyanidin is 460.” is not clear. Please rephrase the sentence. The abbreviation EI% appears for the first time in the text, please define abbreviation in parentheses.
Response 4: The sentence was changed to ‘this formula assumes that the effective mass extinction coefficient for a 1% solution of leucocyanidin (El%) which absorbs at 550 nm in a 1 cm cuvette is 460’(line 160-161).
Point 5: Materials and Methods: page 4, line 166: The abbreviation “RT” appears for the first time in the text, please define abbreviation in parentheses.
Response 5: We have inserted ‘retention time (RT)’ as suggested (line 172).
Point 6: Materials and Methods: page 5, line 193: The abbreviation “OD” appears for the first time in the text, please define abbreviation in parentheses.
Response 6: The word ‘optical density (OD)’was inserted as suggested (line 200).
Point 7: Materials and Methods: page 5, line 202: Please check the word “μmole” in subtitle Ferric reducing antioxidant power (FRAP) assay.
Response 7: This was changed to ‘μmol Fe II equivalents per gram extract’ as suggested (line 209)
Point 8: Materials and Methods: page 6, line 220-221: The abbreviations “LPS” and “IFN-g” appears for the first time in the text, please define abbreviation in parentheses.
Response 8: We have modified to ‘lipopolysaccharide (LPS) and interferon-gamma (IFN-g)’ as suggested (line 228).
Point 9: Results and Discussion: page 8, line 286: The reference in the text “ [34]” is cited wrong please corrected reference in “Weng et al. [34]”.
Response 9: We have corrected to Weng, et al. [34] (line 293).
Point 10: Results and Discussion: page 8, line 287: Check the words “Aspergillus sojae” and write in italics if necessary.
Response 10: Aspergillus sojae was italicized to Aspergillus sojae as suggested (line 295).
Point 11: Results and Discussion: page 8-9, line 312, 335: Check the word “Ellagitannins” and write with small capital letter if necessary.
Response 11: we have changed to lower case ‘ellagitannins’ as suggested (line 320).
Point 12: Results and Discussion: page 8-19, line 320, 324, 325, 326, 398, 400,402…: Check the word “Pyrogallol” and write with small capital letter if necessary.
Response 12: It was changed to lower case ‘pyrogallol’ as suggested (line 330, 334, 335, 336, 412, 414)
Point 13: Results and Discussion: page 9, line 329: Check the words “in vitro” and “in vivo” and write in italics if necessary.
Response 13: Both “in vitro” and “in vivo” were italicized to as suggested (line339).
Point 14: Results and Discussion: page 9, line 334: The abbreviation “ETS” appears for the first time in the text, please define abbreviation in parentheses.
Response 14: ETs was changed to ‘ellagitannins’ as suggested (line 344).
Point 15: Results and Discussion: page 9, line 337: Check the word “Urolithins” and write with small capital letter if necessary.
Response 15: We changed to the lower case ‘urolithins’ as suggested (line 347).
Point 16: Results and Discussion: page 10, line 396: Reference in the text “(Ozturk Sarikaya, 2015)” is incorrectly cited and is missing from the Reference list. Please cite the reference correctly in the Results and Discussion and in the Reference list.
Response 16: We have corrected as suggested and the Cited now as [55], it also has been updated in the reference section too (line 408 and 742-743).
Point 17: Results and Discussion: page 13, line 501: The abbreviation “LAB-” appears for the first time in the text, please define abbreviation in parentheses.
Response 17: We have changed to ‘Lactic acid bacteria (LAB)’as suggested (line 520).
Reviewer 2 Report
The manuscript focuses on the effect of fermentation on the composition of extracts of Miang tea, by non-filamentous and filamentous fungi-based processes. An improvement of polyphenols, and tannins (total and condensed) is generally observed, with respect to to young and mature leaves. This can influence the activities of such extracts, as determined by DPPH and ABTS assays, thus showing increased antioxidant and scavenging activities. A similar increase was also determined by FRAP test.
The authors report an accurate analysis of chemical composition, together with a description of the extraction and separation methods from the plants. The properties are correlated to the chemical composition, and more specifically to the content of polyphenols, tannins, flavonoids. The discussion on these properties are reported in detail, supported by statistical analysis. Some in vivo experiments are also described (effect against induced ROS production in HT-29 cells).
Indeed, as reported by many contributions also of other authors, it is known a strict relationship between the chemical composition and the antioxidant activity (as mentioned by the authors).
I may recommend the publication in Antioxidants after the following revisions:
- Page 9, line 319 and following: as the properties are related to the content of organic compounds (as organic acids) as well as to the chemical structure, I suggest to include some structures of the more relevant organic molecules.
- Page 10, line 396 and following: simila suggestion as before, also to specify number and position of OH groups (which affect the activities of the studied samples).
As a general comment, in my opinion the authors should better underline the advances presented in this contribution, with respect to previous reported results on the same kind of tea extracts (e.g. ref. 8, 9, 13…). Also a comment about the bioactivity found in Miang samples (anti-inflammatory, anticancer, …) with reference to other teas (Oolong, Kombucha, black teas…) could be useful for the readers.
Author Response
Response to Reviewer 2 Comments
We have revised the manuscript following all suggested points of reviewer and all changed points are highlighted with blue color in R1 version. Regarding the questions and suggestions from reviewers, the answers or explanations are responded to one by one as follows;
Point 1: Page 9, line 319 and following: as the properties are related to the content of organic compounds (as organic acids) as well as to the chemical structure, I suggest to include some structures of the more relevant organic molecules.
Response 1: New Figure (Figure 4 in R1 version) has been inserted to address the suggested point (line 326-327).
Point 2: Page 10, line 396 and following: similar suggestion as before, also to specify number and position of OH groups (which affect the activities of the studied samples).
Response 2: New Figure (Figure 4 in R1 version) has been inserted to address the point (line 326-327).
Point 3: As a general comment, in my opinion the authors should better underline the advances presented in this contribution, with respect to previous reported results on the same kind of tea extracts (e.g. ref. 8, 9, 13…). Also a comment about the bioactivity found in Miang samples (anti-inflammatory, anticancer, …) with reference to other teas (Oolong, Kombucha, black teas…) could be useful for the readers.
Response 3: An added paragraph has been included to buttress the findings and do justice to the Results and Discussion section (line 548-580).